# An MT-InSAR Data Partition Strategy for Sentinel-1A/B TOPS Data

**Yuexin Wang** [1], **Guangcai Feng** [1,*], **Zhixiong Feng** [2], **Yuedong Wang** [1], **Xiuhua Wang** [1], **Shuran Luo** [3], **Yinggang Zhao** [1] and **Hao Lu** [1]

1  School of Geosciences and Info-Physics, Central South University, Changsha 410083, China
2  Guangzhou Urban Planning and Design Survey Research Institute, Guangzhou 510060, China
3  Guangdong Research Institute of Water Resources and Hydropower, Guangzhou 510635, China
*  Correspondence: fredgps@csu.edu.cn; Tel.: +86-182-7486-7449

**Abstract:** The Sentinel-1A/B satellite launched by European Space Agency (ESA) in 2014 provides a huge amount of free Terrain Observation by Progressive Scans (TOPS) data with global coverage to the public. The TOPS data have a frame width of 250 km and have been widely used in surface deformation monitoring. However, traditional Multi-Temporal Interferometric Synthetic Aperture Radar (MT-InSAR) methods require large computer memory and time when processing full resolution data with large width and long strips. In addition, they hardly correct atmospheric delays and orbital errors accurately over a large area. In order to solve these problems, this study proposes a data partition strategy based on MT-InSAR methods. We first process the partitioned images over a large area by traditional MT-InSAR method, then stitch the deformation results into a complete deformation result by correcting the offsets of adjacent partitioned images. This strategy is validated in a flat urban area (Changzhou City in Jiangsu province, China), and a mountainous region (Qijiang in Chongqing City, China). Compared with traditional MT-InSAR methods, the precision of the results obtained by the new strategy is improved by about 5% for Changzhou city and about 15% for Qijiang because of its advantage in atmospheric delay correction. Furthermore, the proposed strategy needs much less memory and time than traditional methods. The total time needed by the traditional method is about 20 h, and by the proposed method, is about 8.7 h, when the number of parallel processing is 5 in the Changzhou city case. The time will be further reduced when the number of parallel processes increases.

**Keywords:** MT-InSAR; ground deformation monitoring; Sentinel-1A/B; image partition; block adjustment

## 1. Introduction

Due to large coverage and high-precision, Interferometric Synthetic Aperture Radar (InSAR) has been widely used for mapping surface deformation, such as urban surface deformation [1,2], seismic deformation [3,4], landslide displacement [5–9], and mining subsidence [10]. With the fast development of SAR satellite technology [11], the observation range and frequency are both improved [12–14], providing cycle monitoring for a large-scale or national wide area. However, the traditional processing strategies for Multi-Temporal Interferometric Synthetic Aperture Radar (MT-InSAR) cannot efficiently process the huge number of images with large spatial and temporal coverage. Furthermore, the possible atmospheric phase screen and orbital errors exist in the SAR images with wide spatial coverage are difficult to be corrected. Therefore, optimizing the InSAR processing strategy and parameters is crucial for the application of wide-area InSAR data.

Using supercomputers or distributed computing systems, such as CASearth Cloud Infrastructure Platform [15], and ESA's G-POD environment [16], is a way to improve the data processing efficiency, but it is too expensive to be popularized. Another way is to

segment a large image into small blocks, which can significantly reduce the computation burden and complexity in one block and improve the efficiency of data processing. Currently, data partition strategies are applied in some steps of SAR data processing, such as phase unwrapping [17–20], orbital error correction [21], atmospheric correction [22,23], and PS point decomposition [24,25], but not the whole data process. GAMMA software provides a well-known patch-based point target analysis method, Interferometric Point Target Analysis (IPTA) [26]. However, the method of using local reference points between neighboring patches to extend the results to adjacent blocks is highly affected by unstable connections, resulting in errors propagating in the result easily [27]. StaMPS method and software [28] also provide a block strategy to select permanent scatterers, but it is time-consuming for large-scale areas [29]. Data block processing often introduces systematic errors, such as reference basis errors [29–31]. To remove these errors, external data (GNSS and leveling) and modeling [32,33] are needed. Furthermore, the correction efficacy and precision strongly depend on the precision and spatial distribution of external data.

To address the above problems, we propose a strategy to divide the original data into small blocks by GAMMA software and process these blocks independently by the traditional MT-InSAR method. Then, we use the least square method to estimate the basis between each block and mosaic the corrected block results to obtain the overall results. To validate our strategy, we selected the Sentinel-1 Terrain Observation by Progressive Scans (TOPS) data of a city in the plain area (Changzhou City, Jiangsu Province) and a city in the mountainous area (Qijiang, Chongqing City) in China for the experiment. The results obtained by the traditional and the proposed methods are compared in terms of precision, memory consumption, and time consumption. We also discuss the optimal overlap ratio of blocks and the application of the proposed method.

This study is organized as follows: Section 2 describes the proposed method in detail. The study area and the datasets are introduced in Section 3. In Section 4, we compare the precision and time consumption of the proposed method and traditional method. The block approach and the applicability scenarios of our method is also discussed in Section 5. Finally, some conclusions are drawn in Section 6.

## 2. The Block MT-InSAR Data Processing Strategy

In order to solve the great calculation burden caused by Sentinel-1A/B TOPS data of large spatial and temporal coverage, this paper proposes a data partition strategy based on the MT-InSAR data method, referred to as the block MT-InSAR algorithm. First, the TOPS data are co-registered in the study area to obtain registered single look complex (RSLC), and then the RSLC data are partitioned and processed separately by the traditional MT-InSAR algorithm. Then, the results are corrected by the adjustment model based on the spatial consistency of homonymy points (the same ground deformation points located in different blocks within the overlap areas.). Finally, the results are spliced to obtain the continuous overall deformation results. The general flow of the method is shown in Figures 1 and A1.

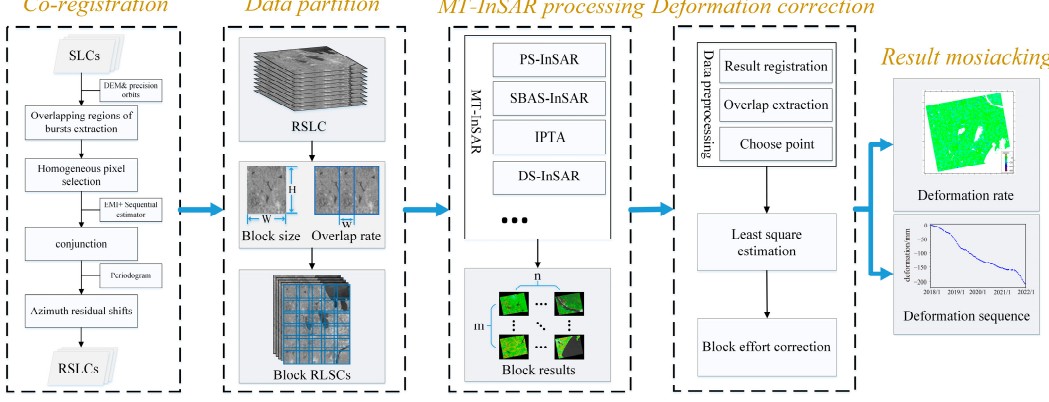

**Figure 1.** Flowchart of the block MT-InSAR algorithm.

### 2.1. Data Partition and Block Processing

Even partition [29], quadtree partition [22], and clustering algorithm partition [19,34] have been used in some parts of data processing, such as atmospheric delay removal, PS network construction, and phase unwrapping. In order to facilitate the splicing process, this paper uses even partition to divide the original data into small blocks, in which the block size and the overlap ratio should be considered.

Block size affects the precision and the processing efficiency of the phase unwrapping, atmospheric delay, and orbit error partition. If the block interferograms are highly coherent and easy to unwrap, the block size has little effect on phase unwrapping precision, but a small block size would lead to high unwrapping efficiency [17]. Additionally, the atmospheric delay in the MT-InSAR consists of a short-scale (few kilometers) and a long-scale (tens of kilometers) component [35], so the block size smaller than these scales is conducive to removing atmospheric delay. However, the too-small block size may remove the long-wavelength deformation signal. The block width and height should be larger than 1/3 ALOS-2 data in range and azimuth for ALOS-2 (70 km) datasets [21]. Therefore, we set the initial block size as ~30 km in length and width for S1A/B TOPS data.

The overlap ratio between blocks also affects the reliability of results and data processing efficiency. The larger the overlap ratio the higher the reliability. For example, if a block area is overlapped with the surrounding blocks in four directions by 10%, 36% of the small block is overlapped with the surrounding blocks, and the overlap area will become 96% when the overlap ratio is 40% in width and height directions. (Figure 2a) However, increasing the overlap ratio will lead to a lot of repeated calculations and reduce the data processing efficiency. In order to improve the result reliability (>50% overlap area), we take off the balance between the reliability of results and choose the overlap ratio of 20~40% for further experiments.

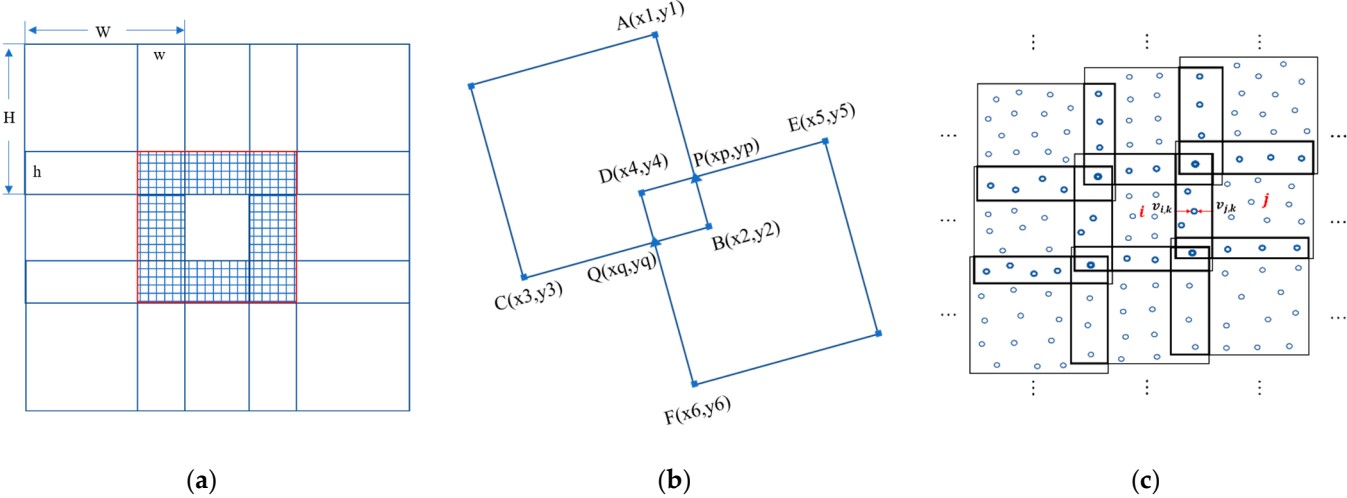

**(a)**    **(b)**    **(c)**

**Figure 2.** (**a**) Diagram of overlap ratio and overlap area. $w$ and $h$ are the width and height of the overlap region, respectively, and $W$ and $H$ are the width and height of the image, respectively. So, the overlap ratio is $w/W$ or $h/H$. The shadow area is the overlap area in the block (red line) with the surrounding blocks, and the overlap area is $1 - (1 - 2 \times w/W) \times (1 - 2 \times h/H)$. (**b**) Diagram of the coordinates acquirement of corner points in the overlap region. (**c**) Diagram of the adjustment model. The circles represent the deformation points. The thicker the circles, the more times the regions are overlapped. $v$ represents the deformation rate of each point.

### 2.2. Results Correction Based on Least Square Estimation

After data partition, we process each block of data to obtain the deformation results using the improved IPTA-InSAR method [36,37]. The obtained deformation results of all the blocks are preprocessed through three steps. (1) Co-registration. Due to the location errors caused by orbital errors and low resolution of DEM; the location of deformation

results may have a systematic deviation of about 1–2 pixels after geocoding. Such deviation can be solved by an overall offset correction using some feature points on the ground. (2) Automatic extraction of overlapping regions. The deformation rates of the homonymy points in the overlapping region determine the correction precision, so it is necessary to identify the overlapping regions between the deformation results first. We use the topological relationship between image overlays (quadrangles) to find the coordinates of corner points in the overlap region (Figure 2b). (3) High-quality homonymy point selection. We select homonymy points with high coherence and small uncertainty. After these operations, the deformation results can be corrected by adjustment.

Errors can be removed during data processing. However, data partition makes each block have a local reference point, and the benchmarks of these reference points may be different, resulting in discrepancies between the results of adjacent blocks and affecting the precision of the overall results. The difference in the deformation rates of the homonymy points in the overlapping area is described as

$$vel_{i,k} - vel_{j,k} = \delta_{i,k} - \delta_{j,k} \tag{1}$$

where $vel_{i,k}$ and $vel_{j,k}$ denote the deformation rate at deformation point $k$ in image $i$ and $j$, respectively, and $\delta_{i,k}$, and $\delta_{j,k}$ denote the error of the corresponding points. Since the error contains mainly the difference in benchmarks, this value can be assumed as a constant.

The matrix form of Equation (1) is:

$$V = B\hat{X} - L \tag{2}$$

where $V = \begin{bmatrix} \delta v_1 & \delta v_2 & \cdots & \delta v_i & \cdots & \delta v_M \end{bmatrix}^T$ is the residual of the calculated values and the observations, $\hat{X} = \begin{bmatrix} \hat{x}_1 & \hat{x}_2 & \cdots & \hat{x}_M \end{bmatrix}^T$ is the difference between the reference points of adjacent images estimated by the least square method. $B$ is the coefficient matrix. $L = \begin{bmatrix} vel_{1,k} - vel_{2,k} & vel_{i,k} - vel_{j,k} & \cdots \end{bmatrix}^T$.

To solve Equation (2), we have to determine the weights of the blocks according to the quality of the data involved in the adjustment.

$$D(L) = \sigma_0^2 P^{-1} \tag{3}$$

$\sigma_0^2$ denotes the variance of unit weight and $P$ is the weight matrix. Assume that the uncertainty of point $i$ is given by $\delta$. Then, the weight of the point is

$$p_i = \frac{C}{\delta_i} \tag{4}$$

In this study, the data are partitioned into small blocks, which are processed independently. The Helmert variance component estimation for multiple data classes is applied to optimize the solution weights of each data set.

$$S\hat{\theta} = W_\theta \tag{5}$$

where $\hat{\theta} = \begin{bmatrix} \hat{\sigma}_{01}^2 & \hat{\sigma}_{02}^2 & \cdots & \hat{\sigma}_{0M}^2 \end{bmatrix}^T$ is the estimated variance of unit weight. $W_\theta$ is the square sum of the corrected values, $W_\theta = \begin{bmatrix} V_1^T P V_1 & V_2^T P V_2 & \cdots & V_M^T P V_M \end{bmatrix}^T$. $S$ is the coefficient matrix. After obtaining $\hat{\theta}$, $\hat{X}$ is solved using the least square method. Repeat the above process until $\hat{\theta}$ satisfies the given threshold $T = 3\delta_0$, and the corresponding solution is the optimal $\hat{X}$ for each SAR image block. The corrected deformation rate is obtained by Equation (6).

$$\hat{vel}_{i,k} = vel_{i,k} - \hat{x}_i \tag{6}$$

The posteriori variance of unit weight and the covariance array are used to evaluate the adjustment observation. They can be obtained by

$$\hat{\sigma}_0^2 = \frac{V^T P V}{r} \tag{7}$$

$$Q_{\hat{x}_i} = B_i^T P B_i \tag{8}$$

In Equation (7), $r$ is the number of redundant observations, and it can be referred to as the number of degrees of freedom, $r = N - M$, with $N$ denoting the number of observations, and $M$ denoting the row number of $\hat{X}$. According to the error propagation, the covariance of the estimate of the homonymy points in the overlap region can be obtained by Equation (9).

$$Q_{\hat{L}\hat{L}} = \left( BQ_{\hat{x}}^{-1}B^T P \right) Q (BQ_{\hat{x}}^{-1}B^T P)^T = BQ_{\hat{x}}^{-1}B^T \tag{9}$$

To verify the precision of the adjustment results, the deformation difference of the homonymy points before and after correction are compared. The block processing results are verified by comparing with that of the traditional processing method (the result without partitioning processing).

### 2.3. Result Mosaicking

The final step is to mosaic the corrected deformation results of all blocks. After geocoding, the block results are horizontally mosaicked. After correction, the deformation of the homonymy points in the overlapping area may still have differences, due to the different errors distribution. We adopt the weighted average method to merge the deformation of the homonymy points.

After correcting the deformation rate, we correct the deformation sequence. Assuming that the deformation is linear, $T(T_1, T_2 \cdots T_a \cdots T_t)$ is a deformation sequence, and the corrected deformation sequence at time $T_a$ can be obtained by

$$\hat{S}_a = \int_{T_1}^{T_a} (v_a + \hat{x}_a) dT = S_a + \hat{x}_a (T_a - T_1) \tag{10}$$

where $\hat{S}_a$ is the accumulated deformation after correction, $v_a$ is the deformation rate. $\hat{x}_a$ is the correction of deformation rate, but cannot be calculated, $S_a$ is the accumulated deformation before correction. If the deformation is linear, $\hat{x}_a$ is equal to the correction of average deformation rate in the deformation sequences $T$. Variant $\hat{x}$ can be calculated by Equation (2), so the equation can be instead of Equation (11). Figure 3 is the diagram of deformation sequence correction.

$$\hat{S}_a = S_a + \hat{x}(T_a - T_1) \tag{11}$$

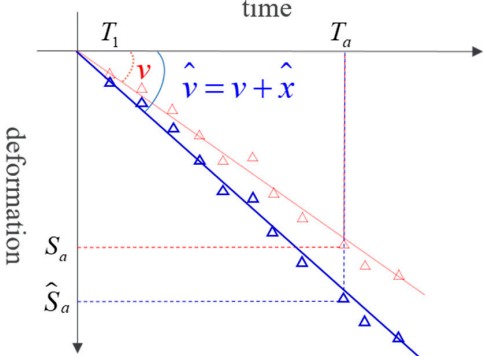

**Figure 3.** Deformation sequence correction diagram.

## 3. Experiment and Data Processing

### 3.1. Study Area and Datasets

Two study areas are selected to validate the proposed method. One is in Changzhou City (31°09′–32°04′N, 119°08′–120°12′E), a coastal city in eastern China. This area is a plain with an elevation of about 10 m [38]. It has a highly developed economy and urban industry. The continuous expansion of urban and engineering projects has changed the geological environment and led to frequent geological hazards. So, surface subsidence monitoring in the area is necessary.

The other study area is Qijiang (28°27′–29°11′N, 106°23′–107°03′E), in western China. It is in the transition zone from the southeastern edge of the Sichuan basin to the Yunnan-Guizhou plateau. The topography is undulating. The mountainous area accounts for 67.6% of the total area and the hills account for 32.4%. The average elevation of this area is 254.8 m [39].

These two regions are used to test the applicability of the proposed method under different error conditions.

We collected 110 Sentinel-1A/B TOPS images covering Changzhou City from path T69 and frame 99, between 5 January 2018 and 31 December 2020, and acquired 115 Sentinel-1 images over Qijiang from path T55 and frame 92, between 9 January 2018 and 31 December 2021. Specific image parameters and image acquisition time are shown in Tables 1 and A1 in Appendix A.

**Table 1.** The image parameters of the study areas.

| Study Area | Direction | Path | Heading | Incidence | Pixel Spacing (Rg × Az) | Num of Images |
|---|---|---|---|---|---|---|
| Changzhou | Ascending | T69 | −12.79° | 36.65° | 2.33 × 13.98 m | 110 |
| Qijiang | | T55 | −12.65° | 43.64° | 2.33 × 13.96 m | 115 |

### 3.2. Data Processing

Using the method described in Section 2, we partitioned the acquired single look complex (SLC) images after co-registration and obtained 30 small blocks with overlapping regions (Figure 4). The block size in Changzhou City is about 7000 × 1400 (pixels), and the overlap rate is about 30%; the block size in Qijiang is about 6400 × 1600 (pixels), and the overlap rate is about 25%.

The spatial baselines of Sentinel-1 images are short, so we connected each image with two (temporally) adjacent images to form a network, only considering the temporal baselines. A multi-look operation (range: azimuth = 5:1) was applied to reduce the noise. After the multi-look operation, the image size of Changzhou city was reduced to 1600 × 1400 (pixels) and that of Qijiang was reduced to 1500 × 1600 (pixels). The data were processed by minimum cost flow (MCF) for phase unwrapping, and Goldstein filtering for noise mitigation. The PS points were selected considering the phase coherence threshold and the amplitude dispersion threshold of the amplitude map. Orbital error phases were removed by polynomial fitting. Most atmospheric phases were removed by differencing between neighboring PS points, and the remaining was removed by spatial-temporal filtering. The topographic residual phases were then removed using linear regression. Finally, the deformation sequence was solved from the remaining phases using Singular Value Decomposition (SVD). The obtained time series of deformation was corrected by the method introduced in Section 2.2. The average deformation of the high-quality homonymy points in the overlapping areas was used for correction. Finally, the result of traditional processing and block processing were obtained. Figure 5 shows the deformation results of Changzhou and Qijiang.

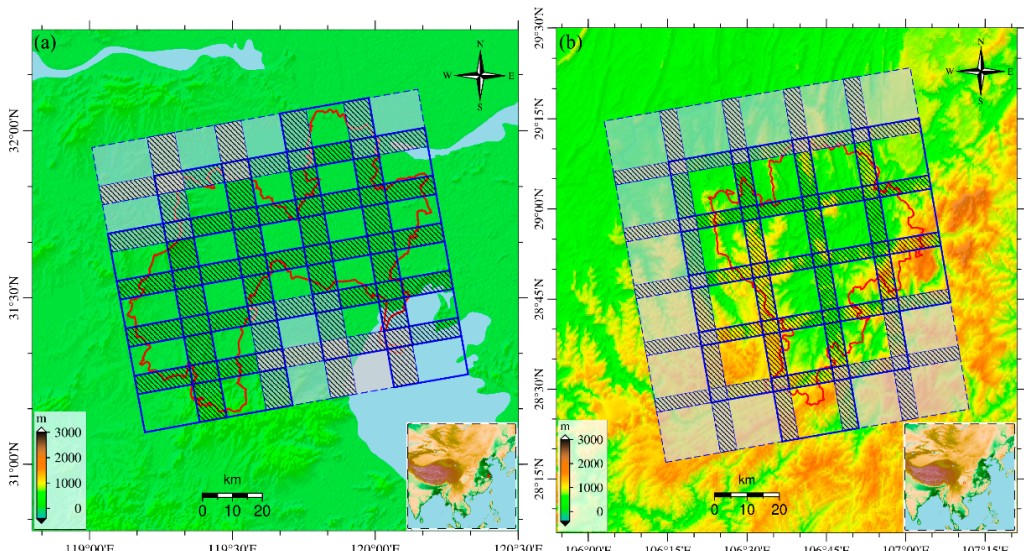

**Figure 4.** Data coverage for (**a**) Changzhou City and (**b**) Qijiang City. The red line is the administrative division boundary. The blue frame is the image coverage after partition and the shaded part is the overlapping area. The gray blocks are not in the study area.

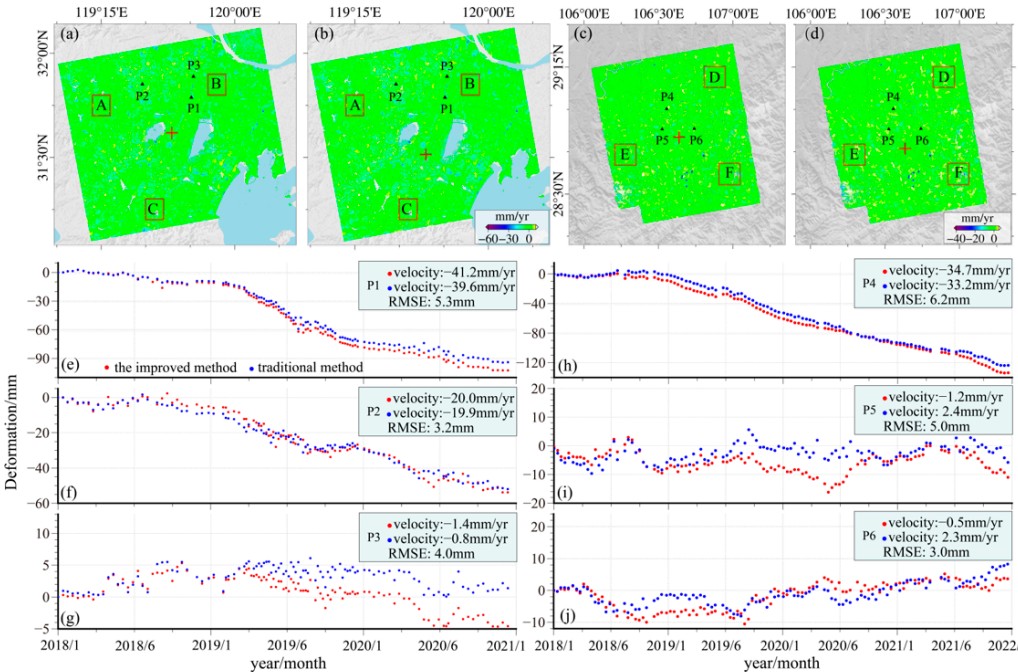

**Figure 5.** Deformation results of the study areas. The results of Changzhou found by (**a**) partition method and (**b**) traditional method. The results of Qijiang found by (**c**) partition method and (**d**) traditional method. (**e–j**) Are time series results of the selected points. The red "+" is the reference point. The reference points in (**a**,**c**) are virtual reference points after free net adjustment because of there are reference points in each small block before adjustment, and they are the center of gravity of the image coverage. The reference points of (**b**,**d**) are the real reference points in data processing.

## 4. Result Analysis

When dealing with the deformation time series of a large area, most conventional algorithms use one reference point for phase unwrapping and solve for PS point deformation rates. If the distance between the PS point and the reference point is large, the precision of the results is low. Reducing the size of image coverage by partition can improve the precision of PS points. However, partition leads to different reference points for different

blocks, so the deformation results should be corrected to follow one benchmark. In this section, the partition and the traditional methods are compared in terms of precision and time consumption.

### 4.1. Precision of the Deformation Rate

We compare the deformation results in Changzhou found by the partition method and the traditional method in Figure 5. The two results show a similar distribution of deformation, but a slight difference in details. We manually selected three regions A, B, and C for analysis. The size of these regions is 1000 × 1000 (pixels) These three regions are stable and outside the deformation region, so we assumed the deformation as 0. The statistical analysis shows that the standard deviation (STD) of the partition results and the traditional results in region A is 3.4 mm/yr and 3.7 mm/yr, respectively, in region B is 4.1 mm/yr and 4.2 mm/yr, respectively, and in region C is 4.6 mm/yr and 4.6 mm/yr, respectively. On the whole, the precision of the results obtained by the partition strategy is slightly higher than that of the traditional processing results.

Figure 5c,d show the deformation results of Qijiang found by the two methods, which generally agreed with each other but some local areas have some slight differences, especially in the circled areas D, E, F (the selection criteria is the same as A, B, C). The traditional results contain a large number of uplift signals, which are not deformation signals but residual errors. These errors are significantly less in the partition results. The STD of the partition results in regions D, E, and F are 3.2 mm/yr, 3.1 mm/yr, and 3.6 mm/yr, respectively, and the correspondence of the traditional results are 3.8 mm/yr, 3.9 mm/yr, and 4.0 mm/yr, respectively. Therefore, the partition method outperforms the traditional method in error removal.

The comparison results in Table 2 show that the partition method has higher precision than the traditional method. In the Changzhou experiment, the former obtained a precision of about 5% higher than the traditional method, and in the Qijiang experiment, the precision improvement is about 15%.

**Table 2.** Precision of the deformation velocity in Changzhou and Qijiang found by the two methods.

| Study Area | Strategy | Area | Number of Points | Std /(mm/yr) | Mean /(mm/yr) | Difference /(mm/yr) | Precision Improvement |
|---|---|---|---|---|---|---|---|
| Changzhou | Partition | A | 564,867 | 3.4 | | | |
| | | B | 589,430 | 4.1 | 4.0 | | |
| | | C | 593,479 | 4.6 | | | |
| | Traditional | A | 561,071 | 3.7 | | 0.2 | 5% |
| | | B | 582,648 | 4.2 | 4.2 | | |
| | | C | 591,199 | 4.6 | | | |
| Qijiang | Partition | D | 992,679 | 3.2 | | | |
| | | E | 916,098 | 3.1 | 3.3 | | |
| | | F | 922,811 | 3.6 | | | |
| | Traditional | D | 986,163 | 3.8 | | 0.6 | 15% |
| | | E | 897,871 | 3.9 | 3.9 | | |
| | | F | 923,017 | 4.0 | | | |

### 4.2. Precision of the Deformation Sequence

After correcting the deformation rates, we corrected the corresponding deformation sequences. We selected the deformation time series of 3 points in each of Changzhou city (P1, P2, P3) and Qijiang (P4, P5, P6) to test the result precision. These points are in the overlapping regions, and they have different deformation magnitudes. P1 and P4 have

large deformation rate, P2 and P5 have medium deformation rate, and P3 and P6 have small deformation rate. The results are shown in Figure 5. In Changzhou City, the difference between the deformation sequences obtained by our method and that obtained by the traditional method is not significant. At P2, the deformation sequences obtained by the two methods almost coincide (Figure 5f), and the deformation rate difference at the three selected points is less than 1 mm/yr. The RMSE between the two results is 5.3 mm for P1, 3.2 mm for P2, and 4.0 mm for P3. We also selected the time series deformation of the three points in Qijiang. The overall deformation trend and deformation magnitude obtained by the two methods are basically the same. The RMSEs between the two results at P4, P5, and P6 are 6.2 mm, 5.0 mm, and 3.0 mm, respectively.

In Figure 5e,h, the annual average deformation rates of these two points are more than 30 mm/yr, the difference between the deformation rates of these two points was about 1.5 mm/yr, and the deformation monitoring precision of InSAR was also basically in this range. A simple proportional function model is used for correcting the time series. If the deformation is nonlinear, the correction of this model might not be appropriate.

### 4.3. Time and Memory Consumption

The partition and parallel processing strategy can reduce the memory consumption of every single process. Additionally, increasing the number of parallel processing can reduce the time consumed by the whole data processing. The partition strategy can be roughly divided into three stages: image partition, MT-InSAR processing, and correction. The total time consumed by partition depends mainly on the processing times of image blocks, that is, the total number of blocks divided by the number of blocks processed in a single parallel session. The traditional processing spends all its time on MT-InSAR processing.

The program running time and memory consumption of the two strategies are listed in Table 3. The traditional method costs about 20 h. The total time required for block processing is 46.6 h when the number of parallel processing is 1. However, when the number of parallel processing is greater than 2, the block processing needs less time than the traditional processing. Additionally, it only needs 8.7 h when the number of parallel processing is 5. In addition, the computer memory occupied by block processing is much lower than that of traditional processing. In this experiment, the single memory occupied by partitioning is only 1/20 of that occupied by the traditional processing. When the image coverage is large or the computer memory is small, the traditional processing may cause memory overflows and the data cannot be processed successfully, but this problem will not happen to our data partition processing.

**Table 3.** Time and memory consumption of traditional processing and block processing with the number of parallel processing 5.

| | Changzhou | | Qijiang | |
| --- | --- | --- | --- | --- |
| | **Traditional** | **Partition** | **Traditional** | **Partition** |
| Original size (pixels) | 29,739 × 6892 | 29,739 × 6892 | 28,104 × 7648 | 28,104 × 7648 |
| Partition strategy | \ | 6 × 5 ~30% overlap | \ | 6 × 5 ~25% overlap |
| Size of block (pixels) | \ | 7147 × 1373 | \ | 6374 × 1574 |
| Platform | CPU: AMD Ryzen 9 5900X 12-Core/RAM:64 G | | | |
| Multi-look | 5:1 | | | |
| Average number of points in a block | \ | 2,941,200 | \ | 2,493,200 |
| Total number of points | 50,941,512 | 51,178,814 | 51,620,120 | 51,883,633 |
| Memory Usage | 27.2 G | 1.3 G | 27.9 G | 1.3 G |
| Time of partition | \ | ~1 h | \ | ~1 h |
| Time of InSAR processing | ~20 h | ~1.2 h | ~20 h | ~1.2 h |
| Time of correction | \ | ~0.1 h | \ | ~0.1 h |
| Total time | ~20 h | ~8.7 h | ~20 h | ~8.7 h |

## 5. Discussion

### 5.1. Space Consistency Correction

The precision of the deformation in each block can be calculated by Formula (9). In the results of Changzhou, the unit weight mean error after adjustment is 0.13 mm/yr, the precision of the block with the highest adjustment precision is 0.27 mm/yr, and the precision of the block with the lowest adjustment precision is 0.59 mm/yr. In the results of Qijiang, the unit weight mean error after adjustment is 0.28 mm/yr, the precision of the block with the highest adjustment precision is 0.40 mm/yr, and the precision of the block with the lowest adjustment precision is 0.81 mm/yr. The adjustment precision is plotted in Figures 6 and 7, which show that the precision of the center blocks is higher than that of the edge blocks.

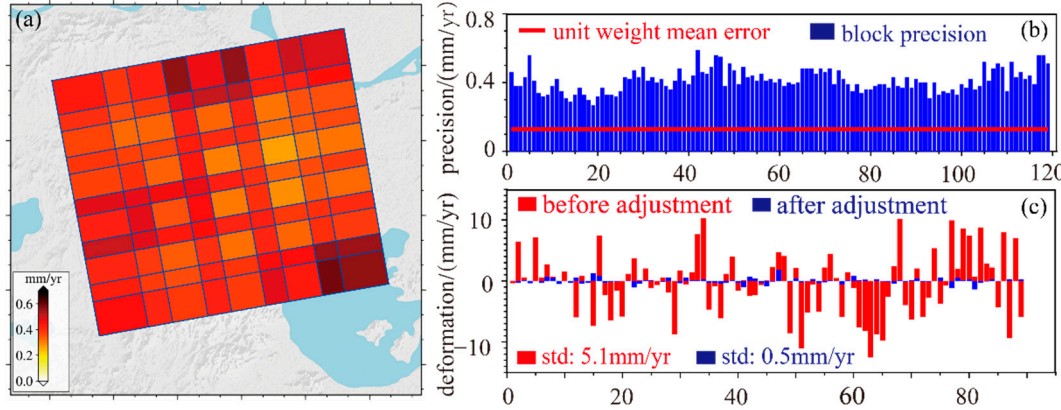

**Figure 6.** (**a**) Deformation precision of each block after adjustment in Changzhou. (**b**) The histogram of the adjustment precision of each block after adjustment, (**c**) the histogram of the difference of the mean value of the overlapping area before and after adjustment of each block.

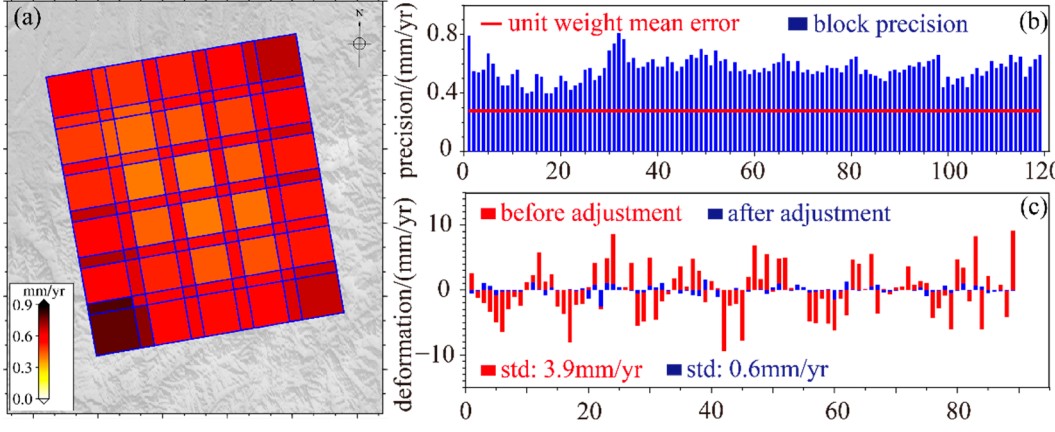

**Figure 7.** The same as Figure 6, but for Qijiang.

In theory, after adjustment, the deformation rates of the homonymy points in the overlapping region should be the same. As Figure 6c shows, before the adjustment, the difference between homonymy points in Changzhou is more than 5 mm/yr, with an STD of 5.1 mm/yr, and after the adjustment, the difference almost converges to 0, with an STD of 0.5 mm/yr. In Qijiang, the difference between homonymy points is more than 7 mm before adjustment, with an STD of 3.9 mm/yr, and it is reduced to 0.6 mm/yr after the adjustment. The precision of the block processing results in the two study areas was greatly improved by adjustment, indicating that adjustment can improve the consistency of the block deformation results.

We selected 4 deformation areas (A, B, C, D) in the overlapping area in Changzhou (Figure A2), and compared their deformation results before and after adjustment in Figure 8. The results of the two image blocks in region A have little difference, so the improvement of the result is not significant after the adjustment (Figure 8b,f). However, the results of regions B, C and D are improved obviously after adjustment. The mean values of the differences of deformation in these three regions change from 4.1 mm/yr, −5.6 mm/yr, and 5.6 mm/yr to 0.3 mm/yr, 0.1 mm/yr, and 0.2 mm/yr after correction, and the spatial consistency of the results improves more significantly.

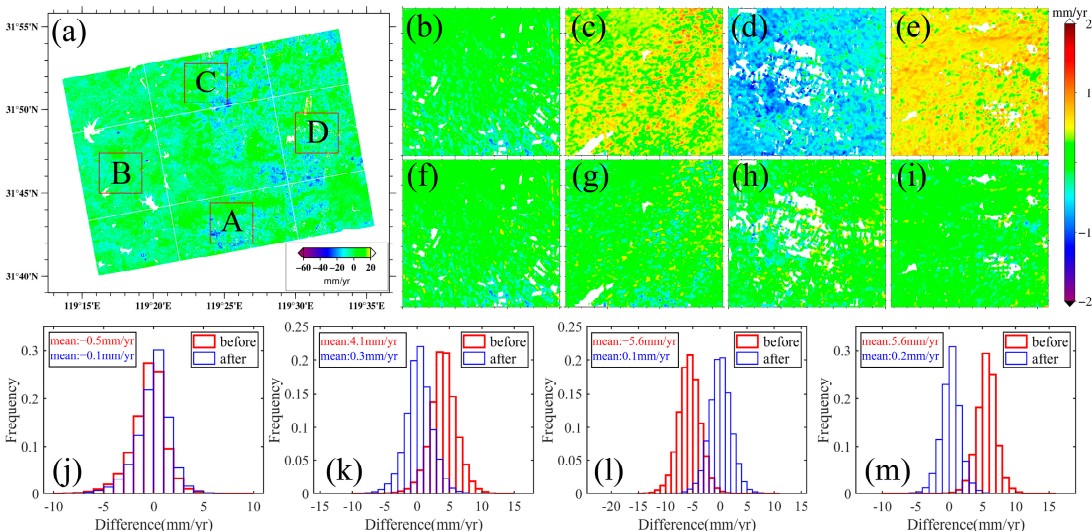

**Figure 8.** The difference in the homonymy points before and after adjustment in regions A, B, C, and D of Changzhou. (**a**) shows the location of the four regions. (**b**–**e**) Are the difference of the overlapping areas before adjustment in regions A, B, C, and D, and (**f**–**i**) are the difference after adjustment. (**j**–**m**) Shows the statistical histograms of the four regions.

### 5.2. Effects of Overlap Rate on Result Precision and Time Consumption

In the proposed partition strategy, partitioning the image is the first and most critical step. Different partition strategies provide different precision results, and different data processing efficiency. Because we adopt the even partition strategy, the size and overlap ratio of the blocks have a great impact on the results. To analyze the effects of the overlap ratio on the partition results, we set one block size and obtained the temporal deformation results of Changzhou using the overlap ratios of 10%, 20%, 30%, and 40%, separately. The results are shown in Figure A2. We evaluated the result precision (Table 4).

The precision of the deformation rates obtained by different overlap ratios is similar because the mean values of the high-quality homonymy points in the overlap region are used in adjustment, which is slightly influenced by the overlap region. As long as the block size is the same, the precision of the results obtained by different overlap ratios are similar. However, the time consumed by different overlap ratios is different. The larger the overlap ratio, the larger the number of blocks and the longer the processing. When the number of parallel processing is 5, the total time consumed by the four overlap ratios are 6.6 h, 7.6 h, 9.5 h, and 10.7 h, indicating that the consuming time increases with the increase of overlap ratio. The increase in the overlapping area brings larger double-counted areas and raises the reliability of the results. Considering the precision, reliability, and time consumption, we choose 20% as the best overlap ratio. The experiments show that overlap ratio 20% has similar result precision and time consumption with that of overlap ratio 10%, but it leads to more than 64% overlap area, which contributes to a significantly higher reliability.

**Table 4.** Results of using different block overlap ratios.

| Overlap Ratio | Total Overlap | Block Size (Amount) | Precision /(mm/yr) | | Time of Each Block | Total Time |
|---|---|---|---|---|---|---|
| Traditional | 0% | 29,739 × 6892 (1) | A<br>B<br>C | 3.7<br>4.2<br>4.6 | 20 h | 20 h |
| 10% | 36% | 7000 × 1500 (20)<br>4539 × 1500 (5) | A1<br>B1<br>C1 | 3.5<br>4.2<br>4.6 | 1.2 h<br>0.8 h | 6.7 h |
| 20% | 64% | 7000 × 1500 (25)<br>7000 × 892 (5) | A2<br>B2<br>C2 | 3.5<br>4.1<br>4.6 | 1.2 h<br>0.7 h | 7.8 h |
| 30% | 84% | 7000 × 1500 (30)<br>5239 × 1500 (6) | A3<br>B3<br>C3 | 3.4<br>4.1<br>4.6 | 1.2 h<br>0.9 h | 10.1 h |
| 40% | 96% | 7000 × 1500 (42)<br>4539 × 1500 (7) | A4<br>B4<br>C4 | 3.3<br>4.0<br>4.5 | 1.2 h<br>0.8 h | 12.7 h |

*5.3. Implications of Data Partition Strategy for MT-InSAR*

The administrative boundary of a city is usually an irregular polygon, but the image coverage is a regular quadrangle. Thus, the image coverage contains many data unrelated to the study area. The traditional method will also process these data. If such data accounts for a large proportion of the image, the data processing will waste a lot of time. The proposed method only processes the block data inside the study area, which can improve the data processing efficiency. In Figure 4, the gray blocks do not need processing. Furthermore, the proposed method can refine the data processing for only the blocks with deformation, which further improves the efficiency of data processing.

The difference in the image coverage will definitely lead to the difference in the results. In this paper, we divide data into blocks, and process, correct, and splice the results of all blocks. The atmospheric delay in small range data is easier to remove than that in the data with large range. Studies have shown that the atmospheric phase in InSAR data measurements has a close correlation with spatial scale [35]. The atmospheric phase difference between two PS points with a distance less than 1 km is less than 0.1 rad2 [40], so the smaller the area, the better the atmospheric error removal according to the error propagation law. However, for large deformation areas, long-wavelength deformation may be removed as orbital errors, due to the polynomial fitting [21]. Thus, the proposed method is not fully applicable to the study area with long-wavelength deformation, such as interseismic deformation.

The correction of the partition results is based on the assumption that the deformation rates of the homonymy points are the same. However, the deformation acquired by InSAR is the line-of-sight (LOS) deformation, and the deformation direction at each point is related to the incidence angle. When the deformation of the homonymy point is obtained from the same orbit and has the same incidence angles, the deformation rates should be the same. If the partitioned data are acquired under different imaging geometries, there will be inconsistency in the incidence angles, resulting in different LOS deformation. Therefore, the incidence angle variation of the results should be considered when the partition data are acquired from different orbits.

Finally, the method does not use control points for the adjustment. Although the benchmarks between image blocks are unified, there may be a deviation between the unified benchmark and the real deformation result datum. We only make a simple correction to the result but using external data as control points may improve the correction.

**6. Conclusions**

In this paper, we propose to partition the data into blocks before obtaining the deformation, to save memory and time for large-scale data processing. To validate this method,

we used the Sentinel-1 TOPS data covering Changzhou, a plain area, and Qijiang, a mountainous area in China. The time series deformation results were obtained in these two regions using the traditional processing method (the improved IPTA) and the partition processing method. The latter outperforms the former in precision, time consumption, and memory occupation. Taking Changzhou City as an example, the memory occupation of the traditional processing method is about 27.2 G, and the total time consumed by processing is about 20 h. During partition processing, the memory occupation of each block is only 1.3 G, and the consumed time is 8.7 h when the parallel number is 5. We also compared the precision of the results obtained by the two methods. The results obtained by the partition processing in Changzhou is as about 4.0 mm/yr, while the precision of the traditional processing is about 4.2 mm/yr. The correspondence in Qijiang is about 3.3 mm/yr and 3.9 mm/yr, respectively. The precision of the results obtained by the proposed method is higher than that obtained by traditional processing.

In general, the proposed method can significantly reduce the memory occupation and time consumption of data processing under the condition of sufficient parallelism, and the precision of the results is higher than that obtained by traditional processing. This method is suitable for monitoring the short-wavelength deformation in a large area, such as large-scale urban deformation monitoring and large-scale landslide deformation detection. However, further research is needed for the result splicing and its application to long-wavelength deformation.

**Author Contributions:** Conceptualization, Y.W. (Yuexin Wang) and G.F.; data curation, S.L.; formal analysis, Z.F. and Y.W. (Yuedong Wang); funding acquisition, Y.W. (Yuexin Wang) and G.F.; methodology, Y.W. (Yuexin Wang) and G.F.; resources, G.F.; Software, Y.W. (Yuexin Wang); supervision, S.L., Y.Z. and H.L.; validation, X.W.; writing—original draft, Y.W. (Yuexin Wang); writing—review and editing, Y.W. (Yuexin Wang), G.F., Z.F., Y.W. (Yuedong Wang), X.W., Y.Z. and H.L. All authors have read and agreed to the published version of the manuscript.

**Funding:** This research was funded by the National Natural Science Foundation of China (No. 42174039), and the Fundamental Research Funds for the Central Universities of Central South University (No. 506021741).

**Acknowledgments:** The authors would like to thank the European Space Agency (ESA) for providing free Sentinel-1 data. The data were additionally retrieved from the Alaska Satellite Facility Distributed Active Archive Center. We also thank the contributors for the Generic Mapping Tools (Wessel et al., 2013) open-source software.

**Conflicts of Interest:** The authors declare no conflict of interest.

## Appendix A

**Table A1.** Parameters of the images used in this study.

| Study Area | Parameters | | Acquisition Date (YYYY/MM/DD) | | | |
|---|---|---|---|---|---|---|
| Changzhou | Direction | Ascending | 2018/01/10 | 2018/01/22 | 2018/02/03 | 2018/02/15 |
| | Path | T69 | 2018/02/27 | 2018/03/11 | 2018/03/23 | 2018/04/04 |
| | Heading | −12.79° | 2018/04/16 | 2018/04/28 | 2018/05/10 | 2018/05/22 |
| | Incidence | 36.65° | 2018/06/03 | 2018/06/15 | 2018/06/27 | 2018/07/09 |
| | Pixel Spacing (Rg × Az) | 2.33 × 13.98 | 2018/07/21 | 2018/08/02 | 2018/08/14 | 2018/09/07 |
| | | | 2018/09/19 | 2018/10/01 | 2018/10/13 | 2018/10/25 |
| | | | 2018/11/06 | 2018/11/18 | 2018/12/12 | 2018/12/24 |
| | | | 2019/01/05 | 2019/01/17 | 2019/02/10 | 2019/02/16 |
| | Number of images | 110 | 2019/02/22 | 2019/03/06 | 2019/03/18 | 2019/03/30 |
| | | | 2019/04/05 | 2019/04/11 | 2019/04/23 | 2019/04/29 |
| | | | 2019/05/05 | 2019/05/11 | 2019/05/17 | 2019/05/23 |
| | | | 2019/05/29 | 2019/06/04 | 2019/06/10 | 2019/06/16 |

**Table A1.** *Cont.*

| Study Area | Parameters | | Acquisition Date (YYYY/MM/DD) | | | |
|---|---|---|---|---|---|---|
| | | | 2019/06/22 | 2019/06/28 | 2019/07/04 | 2019/07/10 |
| | | | 2019/07/16 | 2019/07/22 | 2019/07/28 | 2019/08/03 |
| | | | 2019/08/09 | 2019/08/15 | 2019/08/21 | 2019/08/27 |
| | | | 2019/09/02 | 2019/09/08 | 2019/09/20 | 2019/09/26 |
| | | | 2019/10/02 | 2019/10/08 | 2019/10/14 | 2019/10/20 |
| | | | 2019/10/26 | 2019/11/01 | 2019/11/07 | 2019/11/19 |
| | | | 2019/11/25 | 2019/12/01 | 2019/12/07 | 2019/12/13 |
| | | | 2019/12/19 | 2019/12/25 | 2019/12/31 | 2020/01/12 |
| | | | 2020/01/24 | 2020/02/05 | 2020/02/17 | 2020/02/29 |
| | | | 2020/03/12 | 2020/03/24 | 2020/04/05 | 2020/04/17 |
| | | | 2020/04/29 | 2020/05/11 | 2020/05/23 | 2020/06/04 |
| | | | 2020/06/16 | 2020/06/28 | 2020/07/10 | 2020/07/22 |
| | | | 2020/07/28 | 2020/08/03 | 2020/08/15 | 2020/08/27 |
| | | | 2020/09/08 | 2020/09/20 | 2020/10/02 | 2020/10/14 |
| | | | 2020/10/26 | 2020/11/07 | 2020/11/19 | 2020/12/01 |
| | | | 2020/12/13 | 2020/12/25 | | |
| | Direction | Ascending | 2018/01/09 | 2018/01/21 | 2018/02/02 | 2018/02/14 |
| | Path | T55 | 2018/02/26 | 2018/03/10 | 2018/03/22 | 2018/04/03 |
| | Heading | −12.65° | 2018/04/15 | 2018/04/27 | 2018/05/09 | 2018/05/21 |
| | Incidence | 43.64° | 2018/06/02 | 2018/06/14 | 2018/06/26 | 2018/07/08 |
| | Pixel Spacing | | 2018/07/20 | 2018/08/01 | 2018/08/25 | 2018/09/06 |
| | (Rg × Az) | 2.33 × 13.96 | 2018/09/18 | 2018/09/30 | 2018/10/12 | 2018/10/24 |
| | | | 2018/11/05 | 2018/11/29 | 2018/12/11 | 2018/12/23 |
| | | | 2019/01/04 | 2019/01/16 | 2019/01/28 | 2019/02/09 |
| | | | 2019/02/21 | 2019/03/05 | 2019/03/17 | 2019/03/29 |
| | | | 2019/04/10 | 2019/04/22 | 2019/05/04 | 2019/05/16 |
| | | | 2019/05/28 | 2019/06/09 | 2019/07/03 | 2019/07/15 |
| | | | 2019/07/27 | 2019/08/08 | 2019/08/20 | 2019/09/01 |
| | | | 2019/09/13 | 2019/09/25 | 2019/10/07 | 2019/10/19 |
| | | | 2019/10/31 | 2019/11/12 | 2019/11/24 | 2019/12/06 |
| Qijiang | | | 2019/12/18 | 2019/12/30 | 2020/01/11 | 2020/01/23 |
| | | | 2020/02/04 | 2020/02/16 | 2020/02/28 | 2020/03/11 |
| | | | 2020/03/23 | 2020/04/04 | 2020/04/16 | 2020/04/28 |
| | Number of | 114 | 2020/05/22 | 2020/06/03 | 2020/06/15 | 2020/06/27 |
| | images | | 2020/07/09 | 2020/07/21 | 2020/08/02 | 2020/08/14 |
| | | | 2020/09/07 | 2020/09/19 | 2020/10/01 | 2020/10/13 |
| | | | 2020/10/25 | 2020/11/06 | 2020/11/18 | 2020/11/30 |
| | | | 2020/12/12 | 2020/12/24 | 2021/01/05 | 2021/01/17 |
| | | | 2021/01/29 | 2021/02/10 | 2021/02/22 | 2021/03/06 |
| | | | 2021/03/18 | 2021/03/30 | 2021/04/11 | 2021/04/23 |
| | | | 2021/05/29 | 2021/06/10 | 2021/06/22 | 2021/07/16 |
| | | | 2021/07/28 | 2021/08/09 | 2021/08/21 | 2021/09/02 |
| | | | 2021/09/14 | 2021/09/26 | 2021/10/08 | 2021/10/20 |
| | | | 2021/11/01 | 2021/11/13 | 2021/11/25 | 2021/12/07 |
| | | | 2021/12/19 | 2021/12/31 | | |

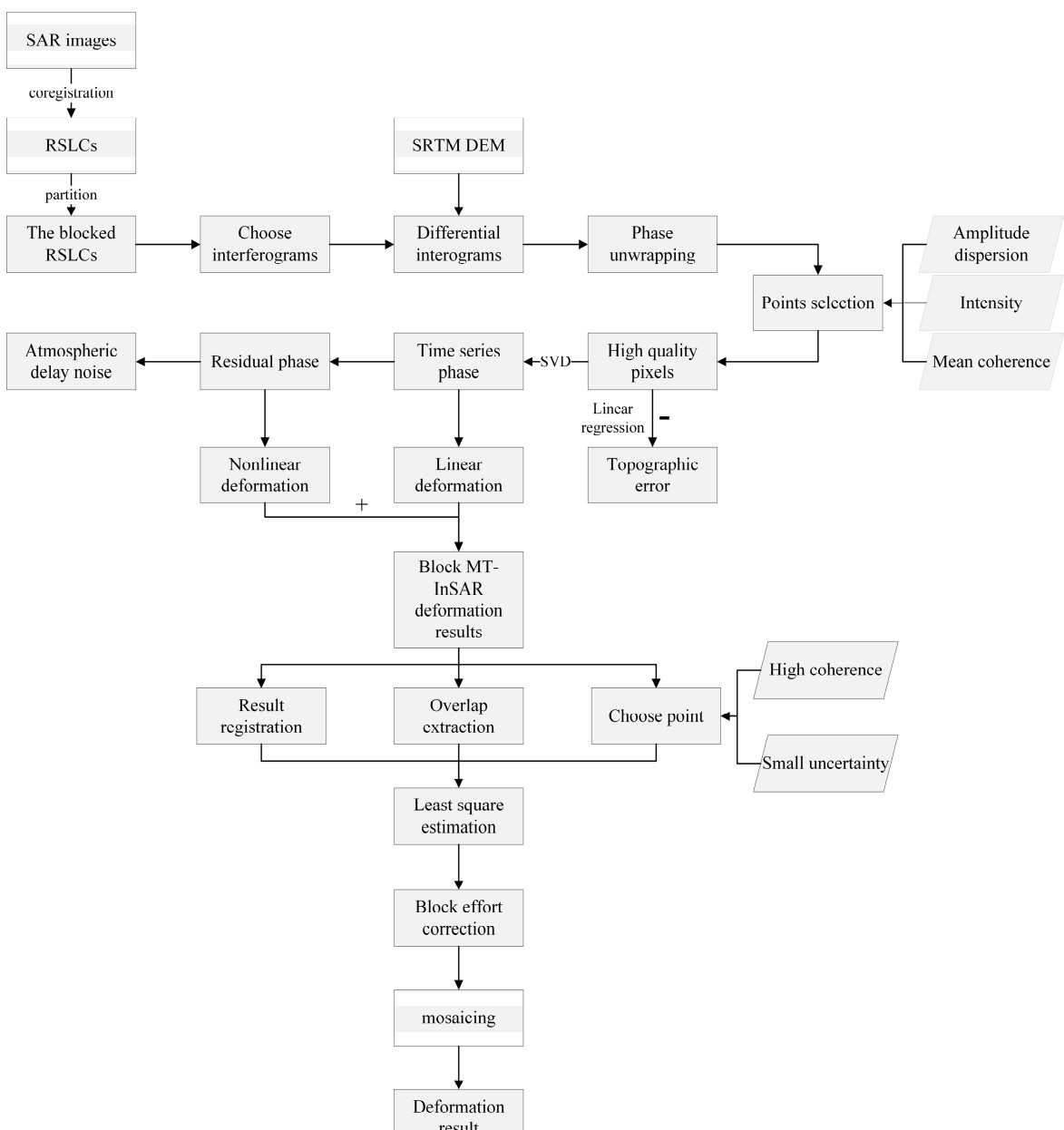

**Figure A1.** The detailed flowchart of the proposed method.

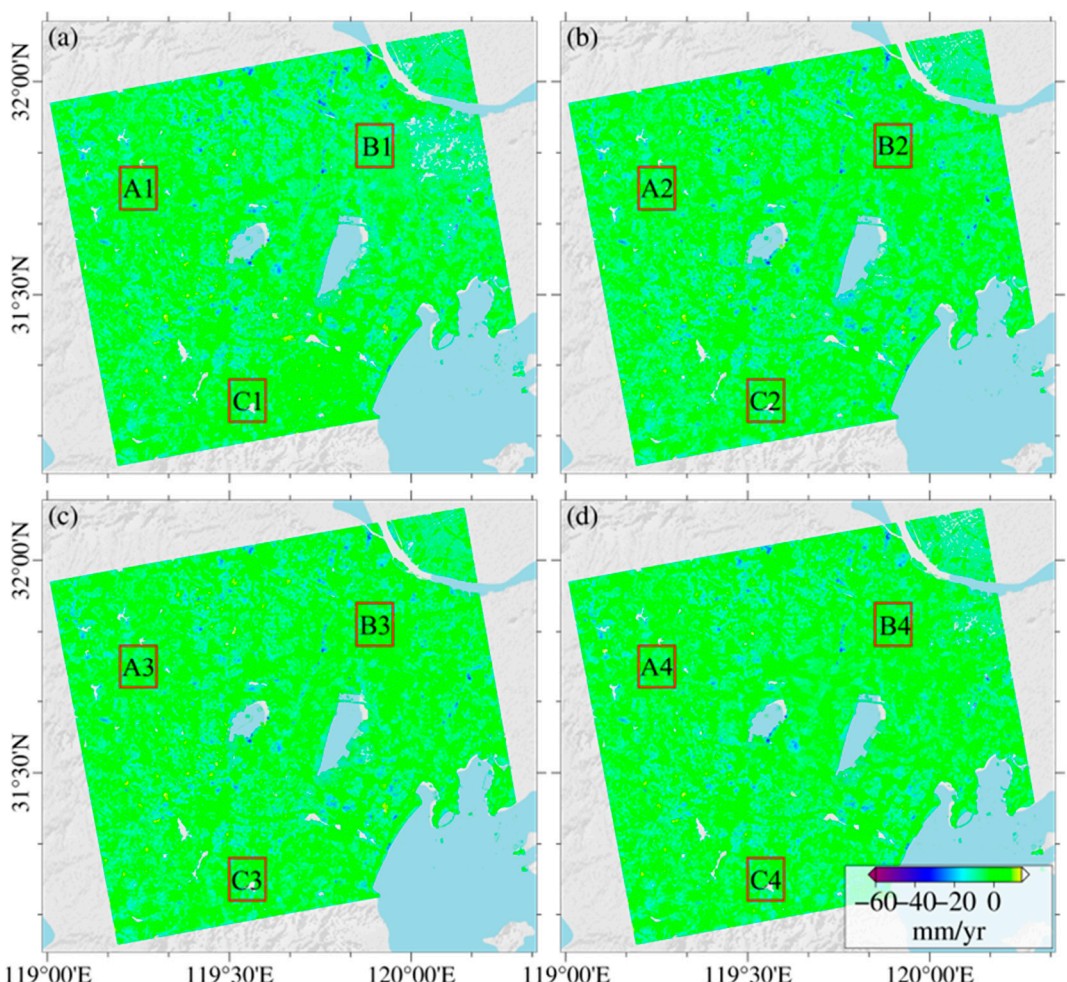

**Figure A2.** The deformation results in Changzhou obtained using the overlap ratio of (**a**) 10%, (**b**) 20%, (**c**) 30%, and (**d**) 40%. A1-C4 are the same areas as A–C described in Section 4.1.

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
