# Peer review of "An MT-InSAR Data Partition Strategy for Sentinel-1A/B TOPS Data"

_remotesensing, doi:10.3390/rs14184562_

Round 1
Reviewer 1 Report
The manuscript entitled "A data partition strategy for MT-InSAR method using for Sentinel 1A/B TOPS data" is a very interesting topic. The authors studied a strategy to reduce time and memory for large scale data processing. In particular, they proposed a procedure based on the partition of satellite images, obtaining interesting results.
The paper is generally well written and structured. However, the proposed manuscript can be accepted as minor revision by making the following suggestions:
In the chapter "introduction" was made a good explanation of the studied phenomena and methodology used. However, I believe the bibliography should be expanded with other recently discussed topics. So, I suggest adding to the literature review the following papers:
1. Novellino, A., et al. "Slow-moving landslide risk assessment combining Machine Learning and InSAR techniques." Catena 203 (2021): 105317.
2. Meng, Qingkai, et al. "Regional recognition and classification of active loess landslides using two-dimensional deformation derived from Sentinel-1 interferometric radar data." Remote Sensing 12.10 (2020): 1541.
3. Miele, Pietro, et al. "SAR data and field surveys combination to update rainfall-induced shallow landslide inventory." Remote Sensing Applications: Society and Environment 26 (2022): 100755.
Data partition methods used in this work should be describe with more details to aid the reader better understand the procedure.
Don't overlapping areas create duplication problems outcomes?
May you provide a logical explanation why the required memory is lower on the same study area using your methodology?
Concerning the time, I can understand why, but on the memory required not really.
You declare that with your methodology the outcomes are more accurate, but based on what? Are the phenomena analysed already known? I do not believe that having lower values means having better results.
In the "conclusions" paragraph, it could be interesting to expand some considerations on the exportability of the method also in other contexts and therefore not to make the work too site-specific.
Check the style citation formats and typos present in the paper.
Reviewer 2 Report
Dear authors, thank you for the manuscript. It regards the comparison of InSAR processing methods: the standard method and a method that divides the AOI into several parts and processes them independently, and then merges the results together.
The paper (including its title) needs to be improved regarding the language.
In case of Sentinel-1 TOPS data, the TOPS post-coregistration step needs to be performed on a large area in order to make the coregistration precise enough. In the paper, I did not find any note about it, and I finally understood that you split the processing after this step. Please make it more clear, if I am right. If not, please specify how you handle this problem.
Please define "homonymy points" to make it clear.
Overlap ratios: "this value will become 96% if the overlap ratio is 40%". Please make it more clear, maybe by defining two different "overlap ratios".
Please define "deformation values" - it is only clear from the following text (and figure 2), where you discuss the time series individually
Eq. (2): please define V
the text between the equations should be also made clearer
Eq (5): "S is the coefficient" - it is not clear what kind of coefficient
please define T
Eq (7): r is the number of redundant observations
I understand that you use IPTA as the "standard processing". What software do you use for the partitioned processing? This information should be given in the introduction
line 226-227: "most conventional algorithms use one reference point for phase unwrapping and PS point selection" - for PS point selection, there is no need of a reference point
selection of regions A-F: please be more clear about the selection criteria. Were they high-quality regions? How far were they from the reference point? And if you present standard deviations (and mean values), could you also present the number of points?
selection of points P1-P6: how the points were selected? please specify also their coherence or RMSE
line 276-277: "if the deformation is nonlinear, the correction effect of this model might not be appropriate" - I think this applies to the case of non-linear deformation rate of the homonymy points
section 4.3: please specify the processing steps requiring large amount of memory. According to me, it is just phase unwrapping... IPTA works by default also in patches...
table 3: please specify also the total number of points
section 5.1: in addition to standard deviations, please specify also the number of points in a block.
line 332-333: are these figures (0.3 mm, 0.1 mm etc.) with regard to the conventional processing, or with regard to the reference point? Should it be 0.3 mm/yr instead? If really mm, what these figures represent?
Reviewer 3 Report
The paper presented a partition deformation extraction method for Sentinel-1A/B SAR images. And relevant experiments show that the proposed method is effective and less time consuming. As a whole, the topic of the submission is interesting. However, due to many reasons mentioned as follows, I have to ask for major revision for the paper:
Comments on the article draft:
[1] The introduction part is deficient and so short, which lacks the review and descriptions of problems, contributions, related methods and related comparisons. An adequate review of these methods should be included in the introduction part.
[2] The comparison methods in the comparison and discussion sections are too few and not up-to-date. Only one method is compared in the result analysis part. I wonder how their results compare to more recent surface deformation methods. This will make the paper results a lot convincing.
[3] What’s ‘traditional method’ in the comparison experiment? Please ensure that it is referenced in the main text.
[4] The detailed algorithm flow graphs of the proposed method are suggested to be added.
[5] The PS method is more reliable for the urban areas than suburb areas. So I wonder, in Figuer 5, maybe it not strange that the precision of Changzhou dataset is superior to Qijiang dataset?
[6] Some variables are not italicized. Such as, Line143, ‘i is given by δ’.
[7] The abbreviation should be defined at first use in the main text. Such as TOPS (terrain observation by progressive scans).
[8] When introducing the applications of InSAR and Sentinel-1A/B in the Introduction Section, some references used here is not representative. Some of them are improper citations, and some are unrelated to Sentinel-1A/B sensors. For example, maybe you should cite the references similar to the following:
1. Ghasemloo, N., Matkan, A.A., Alimohammadi, A. et al. Estimating the Agricultural Farm Soil Moisture Using Spectral Indices of Landsat 8, and Sentinel-1, and Artificial Neural Networks. Journal of Geovisualization & Spatial Analysis, 2022,6(1):19
2. Fan, W., Pan, G. & Wang, L. Development and Application of a Networked Automatic Deformation Monitoring System. Journal of Geovisualization & Spatial Analysis, 2020, 4(1):11
[9] There are lots of English language mistakes and presentation issues (including grammar mistakes and punctuation errors) that have to be thoroughly checked and corrected. For example, line57 ‘are needed, Furthermore’ should be ‘are needed. Furthermore’.
Round 2
Reviewer 2 Report
Dear authors, thank you for the revised manuscript. The language has been significantly improved and requires only minor spellcheck corrections.
Also, most of my comments have been taken into account.
Still, it is not clear what is S in eq(5): coefficient matrix does not provide explanation, and you provided the matrix form only to me, and not to all readers. Could you somehow simply describe from where the coefficient come?
Minor spellchecks to be corrected:
- line 36: high-precision -> high precision
- line 60: large-scales -> large-scale
- line 100: easy unwrapping -> easy to unwrap
- line 102: consists a short scale -> consists OF a short scale
- lines 110, 111: please reformulate the sentence in order to be comprehensible
- figure 2 caption: shadow area -> shadowed area
-line 161: repeat -> repeating
- line 167: r is the NUMBER OF redundant observationS
- line 167: as the NUMBER OF degreeS of freedom
- line 181: after correctING
- line 223: too short -> very short
- line 249: deal -> dealing
- line 279: higher THAN the traditional method
- line 284: after correctING
- line 412: angels -> angles
Reviewer 3 Report
This manuscript proposed a blocking strategy for time-series deformation monitoring, and the authors employ finite overlap and error correction methods in blocking and mosaicking to control the efficiency and accuracy of the proposed method. Overall, the topic, as well as the proposed solution to the presented problem has a scientific soundness. The manuscript would make a litter interesting contribution to the RS journal. However, in its current state, I still have doubts about some details in the text (details below).
1. In the introduction section, it is easier to understand that block processing can improve data processing efficiency, and at the same time, I hope the authors enhance the theoretical support for measurement accuracy improvement.
2. In this paper, the authors demonstrate the effectiveness of the proposed blocking strategy using data from Changzhou and Qijiang, respectively, and use the accuracy correction method in the overlapping area to improve the experimental results in these two study areas compared with the traditional algorithm. However, the authors seem to have overlooked the point that the blocking strategy and block mosaicking should take into account both the characteristics of the data itself and the specific data processing algorithm. For example, in phase unwrapping, although MCF is a local processing algorithm, the establishment of the segmentation line and the selection of the starting point of unwrapping depend on the global processing of the whole image, and if the blocking technique is used to process them separately and then mosaic them, the results will be different from those without blocking, we call this problem " unequal division problem". The mosaicking of the results of the blocking process is dependent on the overlapping region, and if the overlapping region contains more parts with lower data quality, it is difficult to guarantee the quality when mosaicking the results. Of course, I know that this part of improvement does require a larger workload, so I suggest adding two outlooks to the conclusion.
(1) Use adaptive blocking strategy instead of regular blocking strategy to avoid unequal division problem as much as possible.
(2) The data processing algorithm after blocking should use local processing algorithm as much as possible, for example, the region growing phase unwrapping method is more suitable for local processing than MCF
3. After the deformation result is mosaicked, the traces of mosaicking is not very obvious, whether this is due to the filtering of the overlapping parts.
4. References should be given in line 300 (“and the deformation monitoring precision of InSAR was also 300 basically in this range”).
5.The following paper also uses a block approach, which I think the authors can refer to.
“Split-Window Algorithm for Retrieval of Land Surface Temperature Using Landsat 8 Thermal Infrared Data”.
6. The data in this study only consider the ascending data, does it have the same effect for descending data?
7. The authors in the analysis of experimental results, due to the large study area, the analysis focused on the macro level, for the local small range of the two algorithms did not carry out a comparison to illustrate whether there are differences, it is recommended that the authors add local small spatial range comparison analysis content!
8. Is there any difference between the mosaicking effect and the ground truth due to other errors such as noise in the overlapping part of the blocking method?